# Diffusion Coefficient of a Brownian Particle in Equilibrium and Nonequilibrium: Einstein Model and Beyond

**DOI:** 10.3390/e25010042

**Published:** 2022-12-26

**Authors:** Jakub Spiechowicz, Ivan G. Marchenko, Peter Hänggi, Jerzy Łuczka

**Affiliations:** 1Institute of Physics, University of Silesia in Katowice, 41-500 Chorzów, Poland; 2Kharkiv Institute of Physics and Technology, 61108 Kharkiv, Ukraine; 3Education and Research Institute of Computer Physics and Energy, Karazin Kharkiv National University, 61022 Kharkiv, Ukraine; 4Institute of Physics, University of Augsburg, 86135 Augsburg, Germany; 5Max-Planck Institute for Physics of Complex Systems, 01187 Dresden, Germany

**Keywords:** diffusion coefficient, Brownian particle, temperature, Einstein relation, periodic potential

## Abstract

The diffusion of small particles is omnipresent in many processes occurring in nature. As such, it is widely studied and exerted in almost all branches of sciences. It constitutes such a broad and often rather complex subject of exploration that we opt here to narrow our survey to the case of the diffusion coefficient for a Brownian particle that can be modeled in the framework of Langevin dynamics. Our main focus centers on the temperature dependence of the diffusion coefficient for several fundamental models of diverse physical systems. Starting out with diffusion in equilibrium for which the Einstein theory holds, we consider a number of physical situations outside of free Brownian motion and end by surveying nonequilibrium diffusion for a time-periodically driven Brownian particle dwelling randomly in a periodic potential. For this latter situation the diffusion coefficient exhibits an intriguingly non-monotonic dependence on temperature.

## 1. Introduction

In 1784, the Dutch-born British scientist Jan Ingenhousz (best known for his discovery of photosynthesis) described the irregular movement of coal dust on the surface of alcohol [1]. He was not the first who observed the erratic motion of particles, but he detected it for inorganic matter. In 1827, the Scottish botanist Robert Brown described the continuous motion of both organic and inorganic particles in a solution, concluding that the random movement is a general property of matter immersed in the medium [2], and ever since, this phenomenon has been best known as Brownian motion. However, he was not able to explain what he observed, and therefore, Brownian motion did not attract the attention it deserved.

The first good explanation of Brownian movement was developed in 1877 by Joseph Desaulx, who claimed that [3]: *“all the Brownian motions of small masses of gas and of vapour in suspension in liquids, as well as the motions with which viscous granulations and solid particles are animated in the same circumstances, proceed necessarily from the molecular heat motions, universally admitted, in gases and liquids, by the best authorized promoters of the mechanical theory of heat”*. The French physicist and chemist Louis Georges Gouy performed many experiments on Brownian motion from 1888 onwards and found that the magnitude of the motion depends essentially only on two parameters, namely on the size of the particle and the environmental temperature [4]. This work immediately attracted considerable attention of researchers and allowed Brownian motion to be promoted to the rank of one of the most important problems in modern physics. In 1900, the Austrian geophysicist Felix Maria Exner made quantitative studies of the dependence of Brownian motion on the particle size and temperature [5]. He confirmed Gouy’s observation that the movement grows when the size of particle is decreased or temperature is increased. At that time, all research was only qualitative and the theory of Brownian motion was not yet established.

The first correct theoretical description of the diffusion coefficient was provided by William Sutherland in 1904 [6,7] and again in 1905 [8,9], i.e., the celebrated Sutherland–Einstein relation. Salient further details were provided by both Albert Einstein in 1905 [9] and by Marian Smoluchowski in 1906 [10]. While Albert Einstein followed the reasoning analogous to William Sutherland for the diffusion coefficient, more importantly, however, he also included pioneering probabilistic aspects, Marian Smoluchowski’s description was similar in spirit to the use of a kinetic theory via introducing a random walk approach [10]. Einstein’s intention was not to explain Brownian motion because in their 1905 paper he wrote [9]: *“It is possible that the motions to be discussed here are identical with so-called Brownian molecular motion; however, the data available to me on the latter are so imprecise that I could not form a judgment on the question.”* His main objective was to apply the molecular-kinetic theory of heat in order to develop a quantitative theory for the diffusion of small spheres immersed in a suspension and their irregular behavior. In turn, Smoluchowski started their paper by citing Brown’s paper [10]. He introduced a most suitable method, nowadays known as the random walk approach. In 1908, Paul Langevin proposed the third description of Brownian motion based on the Newton equation but additionally complemented with a stochastic force term [11]. This by now is the most widely used approach for various stochastic phenomena. Einstein, Smoluchowski, and Langevin derived an expression for the mean square displacement of the Brownian particle in terms of the corresponding diffusion coefficient. Since that time, the Brownian machinery has started to be used in many areas of science. These original papers opened up a new area of physics, namely the field of statistical physics, when also complemented with underlying fluctuations. Smoluchowski showed how the mathematical apparatus of random walks can be applied in physics, and Langevin introduced stochastic equations for the description of those abundant stochastic phenomena ocurring in nature. All three methods now constitute the foundation of modern statistical physics for both equilibrium and nonequilibrium processes. A historical survey of Brownian motion history is presented with interesting more details in the surveys in References [12,13,14], which are warmly recommended to the interested readership.

In the years that follow up to present times, diffusion phenomena have been studied in the macro- and micro-scale domains, both in the classical regime and in the full quantum regime [15]. Clearly, the phenomenon of diffusion plays an important role not only in physics but also in chemistry, biology, and engineering. Moreover, the theory of diffusion processes is now commonly applied even in sociology, culture, and politics in the context of the spread of ideas, concepts, symbols, knowledge, practices, values, materials, behaviors, and so on [16]. The originators of the theory of Brownian diffusion described only one class of processes, which nowadays are termed *normal diffusion*, i.e., when the spreading of particles trajectories characterizing by the mean-square displacement grows as a linear function of time. If this is not so, then diffusion is termed anomalous, and the corresponding diffusion coefficient cannot be defined in the common way. In this paper, we consider only normal diffusion for which the results by Sutherland and Einstein predict a linear dependence on the environmental temperature for the diffusion coefficient. This feature agrees with our intuition: if this temperature grows higher, then the spread of a cloud of particles should increase as well. However, there are many systems that exhibit deviations from this rule. The diffusion coefficient can show a non-linear function of temperature and, very non-intuitively, even behave non-monotonically with respect to temperature, i.e., it decreases when the temperature increases within some interval. In the next sections, we want to present the simplest systems in which both non-linear and non-monotonic diffusive properties can be observed. Therefore, we will limit our consideration to a one-dimensional dynamics of the classical Brownian particle. Its extension to higher dimensions or even quantum mechanics (being important for the very low-temperature-behavior) will be not be addressed here.

The review is organized as follows. In Section 2, the Brownian motion theory of Einstein [9] is briefly sketched. Moreover, the results published by Sutherland and Smoluchowski are recalled. In Section 3, we introduce the method of a stochastic equation for Brownian motion as pioneered by Langevin in their original paper. In Section 4, a generalized Langevin equation for a Brownian particle is presented. In Section 5, we apply the Langevin approach and briefly rederive the diffusion coefficient for a free Brownian particle when subjected to a constant bias force. Another example is given in Section 6, where we analyze the diffusion coefficient of the particle moving in a spatially periodic potential. Two regimes of the overdamped and underdamped dynamics are addressed. In Section 7, the problem of diffusion in a tilted periodic potential is discussed. Section 8 is devoted to the Brownian particle dynamics moving in a spatially periodic potential while subjected additionally to time-periodic forcing. A discussion and a summary is presented in Section 9. In Appendix A, we detail the scaling scheme for Brownian dynamics and the corresponding dimensionless Langevin equations; our Appendix B contains a derivation of asymptotic behaviors of the diffusion coefficient for the two regimes considered in Section 6. In order to avoid confusion on various notations and symbols used in the original papers, we will use contemporary notation.

## 2. Sutherland–Einstein Diffusion Analysis of Suspended Particles

Einstein, following the reasoning originally put forward by Sutherland in 1904/1905 (see below), applied the molecular-kinetic theory of heat and the Fick relation for the particle flux to describe the diffusion of the solute in a solvent [6,8,9]. Sutherland and Einstein assumed that the only force causing diffusion is given by the gradient of the osmotic pressure *p* and at the dynamic equilibrium, this is related to the Stokes force *F*. If the solute is not too dense, then as far as the osmotic pressure is concerned, it acts as an ideal gas contained in a volume $V^{*}$, and for *n* moles, the relation $pV^{*}=nRT$ holds true. Doing so, a representation of the osmotic pressure in terms of kinetic theory is obtained, reading
(1)$$p=\frac{RT}{N}\rho ,$$
where *R* denotes the gas constant, *T* is the temperature, *N* is the actual number (Loschmidt or Avogadro constant) of molecules contained in a gram-molecule, and ρ is the density of solute molecules, i.e., the number of molecules per unit volume (denoted as ν by Einstein). Hence, the osmotic pressure of a solution depends on the concentration ρ of dissolved solute particles. If a concentration of solute varies in space, the diffusion flux J=ρv (*v* is the particles velocity) is described by Fick’s law, i.e.,
(2)$$J=\rho v=-D\frac{\partial \rho }{\partial x},$$
where *D* is the diffusion coefficient of the suspended substance. He derived a condition for the dynamic equilibrium, namely,
(3)$$F\rho =-\gamma v\rho =\frac{\partial p}{\partial x},$$
where the Stokes force is given by F=−γv, wherein for the spherical particles of radius *a* (denoted as P by Einstein), the Stokes friction γ is given by γ=6πηa, with η denoting the viscosity (labeled as k by Einstein). A differentiation of Equation (Equation 1) with respect to *x* yields
(4)$$\frac{\partial p}{\partial x}=\frac{RT}{N}\frac{\partial \rho }{\partial x}.$$ From Equations (2)–(4), it follows that [6,8,9]
(5)$$D=\frac{RT}{N}\frac{1}{6\pi \eta a}.$$ In modern notation, this central result is recast differently, introducing the Boltzmann constant $k_{B}$, the gas constant per molecule $k_{B}=R/N$, and the friction coefficient γ=6πηa, transforms this diffusion coefficient into its appealing form, reading
(6)$$D:=D_{0}=\frac{k_{B}T}{\gamma }.$$ The diffusion coefficient typically depends on several parameters such as the temperature and size of the particle. In 1877, J. Desaulx stated that [3]: *“The Brownian motion is more active in heated liquids than in those of a low temperature”.* Next, L. G. Gouy wrote that motion decreases with the viscosity of the fluid and identified the randomness with thermal motion [4]. He also observed that the Brownian movements appear more swift for particles of smaller size. F. M. Exner made similar conclusions on those properties of Brownian motion [5]. All these observations are consistent with the result given by Equation (Equation 5).

In the second part of his paper, Einstein used probability theory to derive the diffusion equation for the probability distribution of the suspended particles. There, two important results are derived: (i) the diffusion coefficient is defined via the relation of the mean square displacement by the second central moment (see the first equation on page 558 in [9]) and (ii) the expression of the mean square displacement is given by the salient expression
(7)$$\langle x^{2}\rangle =2Dt.$$ This formula was indeed new, and it opened a new field in which statistical methods were applied to model stochastic motion of particles. In two subsequent papers [17,18], Einstein suggested a way to corroborate experimentally their theory. The French physicist Jean Baptiste Perrin conducted a series of experiments that confirmed Einstein’s predictions [19]: *“I am going to summarize leaving no doubt of the rigorous exactitude of the formula proposed by Einstein”.* In 1926, Perrin was awarded the Nobel Prize for his works on Brownian motion.

### Sutherland’s and Smoluchowski’s Approach

Einstein’s first paper on diffusion was from 11 May 1905. However, Sutherland already in June 1904 communicated results for the diffusion coefficient at the meeting of the *Australian Association for the Advancement of Science*, which took place at Dunedin, New Zealand, (Reference [6]); see also the review article entitled “Correcting the error: Priority and the Einstein papers on Brownain motion” by Boardman [7]. Sutherland’s 1904 results were published again in March 1905 in Phil. Mag. [8], i.e., 2 months before Einstein’s 1905 paper received on May 11. His pioneering reasoning is detailed in Equations (Equation 2)–(Equation 5) therein. Sutherland’s expression even presents a generalization over the Einstein formula by using the more general result for the Stokes friction force *F* [6,8], i.e.,
(8)$$F=-6\pi \eta av\;\frac{1+2\eta /\lambda a}{1+3\eta /\lambda a}$$
with λ denoting the coefficient of sliding friction between the diffusing Brownian particle and the solvent, originally denoted by Sutherland as β. Consequently, Sutherland’s more general result for the diffusion coefficient reads explicitly
(9)$$D=\frac{RT}{N}\frac{1}{6\pi \eta a}\frac{1+3\eta /\lambda a}{1+2\eta /\lambda a}.$$ One observes that for λ→∞, i.e., yielding *zero slip* between the Brownian particle and the surface of the dilute solution molecules, this more general expression for the diffusion coefficient *D* is reduced to the result by Einstein. This result applies to the case of a sizable Brownian particle in the solute, undergoing Brownian trembling movements among those much smaller molecules of the solvent. In the opposite limit of full slip (i.e., for λ→0), which would be realized for a gas-bubble-Brownian particle, the Stokes/Einstein factor of 6 is changed to a smaller value of 4, cf. Equation (Equation 9). These two limiting results are detailed in their Equation (Equation 4) in [8].

In September 1906, Marian Smoluchowski published a famous paper on Brownian motion. He applied a quite different method, which is based on a random walk concept and obtained the formula (see Equation (Equation 3) in *§* 21 in [10])
(10)$$D=\frac{32}{243}\;\frac{\langle mv^{2}\rangle }{\pi \eta a}.$$ Using the equipartition theorem for the kinetic energy $\langle mv^{2}\rangle =RT/N=k_{B}T$, one obtains the form
(11)$$D=\frac{64}{81}\frac{RT}{N}\frac{1}{6\pi \eta a}.$$ For the case in three dimensions (as considered by Smoluchowski), the result should be multiplied by a factor of 3 (i.e., $\langle mv^{2}\rangle =3k_{B}T$), yielding the prefactor, which is larger by a factor of 64/27 than Einstein’s factor 1/6. He, however, realized later that for liquids, their approach precisely recovers the 1/6 by Einstein; see the discussion in Reference [20]. This difference in factors is because Smoluchowski assumed a situation of diffusion in a gas in three dimensions, where the size of the Brownian particles is much smaller than the mean free path of the surrounding gas molecules. This is in distinct contrast to the situation with a liquid solution made up of molecules possessing a much smaller mean free path compared to the size of the suspended Brownian particle. The physical origin of the feature for the two different numerical factors between Einstein’s and Smoluchowski’s expressions is presumably not well known in the present-day community.

A great contribution was made by Smoluchowski, who for the first time introduced the methodology nowadays known as the random walk approach. In his original paper, however, he did not use this random walk terminology. This notion was first used in 1905, one year before Smoluchowski’s paper, by K. Pearson [21].

## 3. Langevin Equation

The Langevin description of diffusion of the Brownian particle of mass *m* in fluid is based on the following Newton Equation [11],
(12)$$m\frac{d^{2}x}{dt^{2}}=-\gamma \frac{dx}{dt}+X,$$
where originally in their paper, the friction coefficient was written as γ=6πμa. He wrote, *“About the complementary force X, we know that it is indifferently positive and negative and that its magnitude is such that it maintains the agitation of the particle, which the viscous resistance would stop without it”*. Today, the force *X* is called thermal white noise and models the influence of the environment (thermostat) of temperature *T* on the Brownian particle. Starting from this stochastic equation, Langevin derived an equation for $x^{2}$, namely,
(13)$$\frac{m}{2}\frac{d^{2}x^{2}}{dt^{2}}=mv^{2}-\frac{1}{2}\gamma \frac{dx^{2}}{dt}+xX,$$
where $v=\dot{x}=dx/dt$ is the velocity of the Brownian particle. In the next step, one can perform the average of both sides of this equation and then apply the energy equipartition theorem $\langle mv^{2}/2\rangle =RT/2N$ and note that 〈xX〉=〈x〉〈X〉=0. The emerging result reads
(14)$$m\frac{dz}{dt}=-\gamma z+2\frac{RT}{N},\;z=\frac{d}{dt}\langle x^{2}\rangle .$$ The solution of this linear differential equations reads
(15)$$z\left(t\right)=\frac{d}{dt}\langle x^{2}\rangle =2\frac{RT}{\gamma N}+C_{0}e^{-t/\tau_{0}},\;\tau_{0}=\frac{m}{\gamma },$$
where $C_{0}$ is the integration constant and $\tau_{0}$ is the relaxation time of the particle velocity. Integration over time gives the mean square of displacement
(16)$$\langle x^{2}\rangle -\langle x_{0}^{2}\rangle =2\frac{RT}{\gamma N}t+C_{1}\left(1-e^{-(\gamma /m)t}\right)$$
which for long times t≫m/γ yields the relation
(17)$$\langle x^{2}\rangle -\langle x_{0}^{2}\rangle ∼2\frac{RT}{\gamma N}t=2Dt.$$ Comparison of this equation with Equation (Equation 5) allows one to obtain the diffusion coefficient *D*, which has the same form as Einstein’s result in Equation (Equation 4), i.e., $D=D_{0}=k_{B}T/\gamma$.

One must note that in the starting Langevin Equation (Equation 12), the Brownian particle is characterized by its mass *m*, which also appears in the full solution (Equation 16). However, in the long time limit, neither the mean square displacement (Equation 17) nor the diffusion coefficient $D_{0}$ depends on the Brownian particle mass *m*.

## 4. Generalized Langevin Equation

In modern statistical physics, generalizations of the Langevin equation are one of the most important methods for the analysis of equilibrium and non-equilibrium phenomena. For a classical particle moving in a potential U(x) and subjected to the deterministic or stochastic force F(t), it takes the form
(18)$$m\ddot{x}\left(t\right)+\int_{0}^{t}\Gamma \left(t-s\right)\dot{x}\left(s\right)ds=-U^{\prime }\left(x\left(t\right)\right)+F\left(t\right)+\mu \left(t\right),$$
being known as the *Generalized Langevin Equation* (GLE) [13,15,22,23,24]. The dot and prime denote differentiation with respect to the time and the particle coordinates, respectively. This GLE is an integro-differential equation possessing the random noise term, and an integral term, which describes the memory effects associated with the interaction of the Brownian particle with the environment (thermostat) of temperature *T*. The memory function Γ(t) decays for a long time and mimics the dissipation mechanism. The random term μ(t) models the thermal equilibrium fluctuations, and according to the fluctuation-dissipation theorem, it satisfies the relations [22,23,24]
(19)$$\langle \mu \left(t\right)\rangle =0,\;\langle \mu \left(t\right)\mu \left(s\right)\rangle =k_{B}T\;\Gamma \left(t-s\right).$$ In a general case, the noise μ(t) is correlated with a non-zero correlation time $\tau_{c}$, and therefore, it is often called colored noise [24,25]. As an example, we can mention one of the simplest memory functions
(20)$$\Gamma \left(t\right)=\frac{\gamma }{\tau_{c}}e^{-\left|t\right|/\tau_{c}}.$$ From Equation (Equation 19), it follows that under such a choice, μ(t) is exponentially correlated and modeled by the Ornstein–Uhlenbeck process. When the correlation time $\tau_{c}$ tends to zero, the noise correlation function is
(21)$$\langle \mu \left(t\right)\mu \left(s\right)\rangle =2\gamma k_{B}T\;\delta \left(t-s\right)$$
and Equation (Equation 18) reduces to the stochastic differential equation reading
(22)$$m\ddot{x}+\gamma \dot{x}=-U^{\prime }\left(x\right)+F\left(t\right)+\sqrt{2\gamma k_{B}T}\xi \left(t\right),$$
where we introduced the rescaled Gaussian thermal noise ξ(t) with characteristics
(23)〈ξ(t)〉=0,〈ξ(t)ξ(s)〉=δ(t−s). The form (Equation 22) with the prefactor $\sqrt{2\gamma k_{B}T}$ is more convenient because it explicitly realizes the fluctuation-dissipation theorem [22,26,27,28]: If there is no dissipation, i.e., γ=0, there are no fluctuations and the noise term vanishes in the same way.

The diffusion process is nowadays characterized by the coordinate variance
(24)$$\langle \Delta x^{2}\left(t\right)\rangle =\langle {\left[x\left(t\right)-\langle x\left(t\right)\rangle \right]}^{2}\rangle =\langle x^{2}\left(t\right)\rangle -{\langle x\left(t\right)\rangle }^{2}=2Dt,$$
which defines normal diffusion as well as its coefficient. If the variance is not a linear function of time, the diffusion is termed anomalous. In many cases the relation (Equation 24) is fulfilled in the long time regime, where a normal diffusion regime is observed. In the sections that follow, we consider such situations.

## 5. Diffusion under a Constant Force

We start with the simplest case of a Brownian motion when U(x)=0 and the Brownian particle is subjected solely to a constant force $F\left(t\right)=F_{0}$. Then Equation (Equation 22) reduces to the form
(25)$$m\ddot{x}+\gamma \dot{x}=F_{0}+\sqrt{2\gamma k_{B}T}\xi \left(t\right).$$ It is still a linear stochastic differential equation which can be solved by standard methods. Another approach is based on the corresponding Fokker-Planck equation which is also tractable analytically. The third method is the simplest and is based on the transformation of Equation (Equation 25). Let
(26)$$y\left(t\right)=x\left(t\right)-\frac{F_{0}t}{\gamma }$$ The new process y(t) is determined by the Langevin equation
(27)$$m\ddot{y}+\gamma \dot{y}=\sqrt{2\gamma k_{B}T}\;\xi \left(t\right),$$
which has a similar form as the original Langevin Equation (Equation 12). Therefore, at the long times we find that
(28)$$\langle {\left[x\left(t\right)-\langle x\left(t\right)\rangle \right]}^{2}\rangle =2\frac{k_{B}T}{\gamma }t=2D_{0}t\;,$$
and the diffusion coefficient $D:=D_{0}$ has the same form as the Einstein one. In the above expression the term $\langle x\left(t\right)\rangle =F_{0}t/\gamma$ is necessary because in contrast to the case (Equation 12) the longtime limit is non-zero and consequently the average velocity of the particle is finite.

## 6. Diffusion in Spatially Periodic Potentials

As a next example of a diffusion process we consider the stochastic dynamics of an inertial Brownian particle of mass *m* moving in a spatially periodic potential U(x)=U(x+L) of period *L*. This setup is described by the following dimensional Langevin dynamics
(29)$$m\ddot{x}+\gamma \dot{x}=-U^{\prime }\left(x\right)+\sqrt{2\gamma k_{B}T}\;\xi \left(t\right).$$ This class of systems is the most important as it models the stochastic particle transport of a plentiful number of relevant practical applications in various fields of science and technology. Equation (Equation 29) describes pendulums [29], super-ionic conductors [30], stochastic dynamics in Josephson junctions [27,31,32], dipoles rotating in external fields [33], phase-locked loops [34], dislocations in solid state physics [35], solitons described by the Sine-Gordon Equation [36], the Frenkel-Kontorova lattices [37], dynamics of adatoms [38], charge density waves [39] and cold atoms in optical lattices [40], to name but a few.

In a periodic potential, the particle performs a random walk between the local minima of the potential. The conservative force $F\left(x\right)=-U^{\prime }\left(x\right)$ attempts to pull it down towards the minimum, while thermal fluctuations randomly agitate the particle for its escape to the left or to the right direction. Eventually, this system reaches its stationary equilibrium state. Consequently, according to the second law of thermodynamics, the averaged velocity vanishes identically, independently of the form or symmetry of the potential U(x). In the long time regime, the mean square displacement obeys the relation (Equation 24) and diffusion is normal. The analytical expression for the diffusion coefficient is not known in a general case; however, there are two limiting regimes for which it has been derived.

### 6.1. Overdamped Dynamics

The first limiting regime corresponds to the Smoluchowski (i.e., overdamped) dynamics for which the inertial effects can be neglected. Formally, the mass is zero, m=0, and the Langevin equation then reads
(30)$$\gamma \dot{x}=-U^{\prime }\left(x\right)+\sqrt{2\gamma k_{B}T}\;\xi \left(t\right).$$ In this case, the problem has been solved by Lifson and Jackson [41], who obtained the diffusion coefficient in a closed form, reading
(31)$$D=\frac{D_{0}}{E\left[e^{-U/k_{B}T}\right]\;E\left[e^{U/k_{B}T}\right]},$$
where $D_{0}$ is the Einstein diffusion coefficient (Equation 4) and E indicates the average over the spatial period *L* of the potential, namely, for any function G(x)
(32)$$E\left[G\right]=\frac{1}{L}\int_{0}^{L}G\left(x\right)dx.$$ From Equation (Equation 31), it follows that in a periodic potential the diffusion coefficient is always smaller than the free diffusion constant, $D\leq D_{0}$. An interesting feature is that the diffusion coefficient *D* remains unchanged if the potential is reversed in sign, U(x)→−U(x). The above formula has been rederived by other authors, e.g., in Reference [42].

As an example, let us choose the generic periodic potential
(33)$$U\left(x\right)=-d_{0}\cos\left(x\right).$$ In this case, the diffusion coefficient takes the result
(34)$$D=D_{1}=\frac{D_{0}}{I_{0}^{2}\left(d_{0}/k_{B}T\right)}=\frac{k_{B}T/\gamma }{I_{0}^{2}\left(d_{0}/k_{B}T\right)},$$
where $I_{0}\left(x\right)$ is the modified Bessel function of the first kind [43]. Let us apply the asymptotic behavior $I_{0}\left(x\right)\approx exp\left(x\right)/\sqrt{2\pi x}$, valid for large values of *x*. Then for low temperatures $k_{B}T\ll d_{0}$, the diffusion coefficient *D* can be approximated by the expression
(35)$$D=2\pi \frac{d_{0}}{\gamma }\;e^{-2d_{0}/k_{B}T}\to 0\;for\;T\to 0.$$ This shows how *D* approaches zero when temperature tends to zero. It should be contrasted with the linear decrease in the Einstein diffusion coefficient $D_{0}\propto T$. In the opposite limit of high temperature, when $k_{B}T\gg d_{0}$, it approaches Einstein’s one, $D\to D_{0}$. In Figure 1, we depict the dependence of *D* given in Equation (Equation 34) *vs.* temperature *T*.

### 6.2. Underdamped Dynamics

The second limiting regime corresponds to the underdamped dynamics. It is convenient to work with scaling defined in Equation (Equation 61) in Appendix A for which the Langevin Equation (Equation 29) assumes the dimensionless form [44]
(36)$$\ddot{x}+\Gamma \dot{x}=-V^{\prime }\left(x\right)+\sqrt{2\Gamma \theta }\;\xi \left(t\right).$$ Details and all dimensional and dimensionless quantities are defined in Equations (Equation 61)–(Equation 65). In this scaling, the dimensionless mass equals M=1, the spatial period of the potential V(x)=V(x+2π) is L=2π, and the rescaled temperature is $\theta =k_{B}T/\Delta U$, where ΔU is half of the potential barrier of the dimensional potential U(x)=U(x+L). The underdamped regime corresponds to the case when the dimensionless friction coefficient Γ≪1; see Chapter 11.4 in [45]. Then, the energy
(37)$$\frac{v^{2}}{2}+V\left(x\right)=E$$
is a slowly varying function of time. The next step is to change the phase space of the system (x,v)→(x,E) with
(38)$$v\left(x,E\right)=+\sqrt{2(E-V(x))}$$
and analyze the problem in the (x,E)-variables. The calculation of the diffusion coefficient is a non-trivial task. However, a systematic and rigorous method has been presented in Reference [44]. The result reads
(39)$$D^{*}=\frac{8\pi^{2}\theta }{\Gamma Z_{\theta }}\int_{E_{0}}^{\infty }\frac{e^{-E/\theta }}{S(E)}dE,$$
where $D^{*}$ is the dimensionless diffusion coefficient, $E_{0}$ is the maximum of the potential, $E_{0}=max\left[V\left(x\right)\right]$, the partition function is
(40)$$Z_{\theta }=\sqrt{2\pi \theta }\;\int_{0}^{2\pi }e^{-V(x)/\theta }dx$$
and
(41)$$S\left(E\right)=\int_{x_{1}\left(E\right)}^{x_{2}\left(E\right)}v\left(x,E\right)dx.$$ Two values $x_{1}\left(E\right)$ and $x_{2}\left(E\right)>x_{1}\left(E\right)$ are determined from the equations
(42)$$\begin{array}{c} V\left(x_{i}\right)=E,\;\;i=1,2,\;for\;E<E_{0} \end{array}$$
(43)$$\begin{array}{c} x_{1}\left(E\right)=-\pi ,\;x_{2}\left(E\right)=\pi ,\;for\;E>E_{0}. \end{array}$$ Let us stress that the formula in Equation (Equation 39) is exact in the limit Γ→0 and also is valid for all periodic potentials and temperatures. In a particular case of the potential
(44)$$U\left(x\right)=-d_{0}\cos\left(x\right),\;i.e.\;V\left(x\right)=-\cos\left(x\right)$$
the dimensionless diffusion coefficient $D^{*}$ takes the appealing form
(45)$$D^{*}=\frac{\sqrt{\pi \theta }}{2\Gamma \;I_{0}\left(1/\theta \right)}\int_{1}^{\infty }\frac{e^{-E/\theta }}{\sqrt{E+1}\;E\left(\sqrt{2/(E+1)}\right)}dE,$$
where
(46)$$E\left(k\right)=\int_{0}^{\pi /2}\sqrt{1-k^{2}\sin^{2}\varphi }\;d\varphi ,\;0\leq k\leq 1,$$
is the complete elliptic integral of the second kind [43]. In dimensional form, i.e., for the process defined by the dimensional Equation (Equation 29), the diffusion coefficient *D* reads
(47)$$D=D_{2}=D_{0}\;\frac{\sqrt{\pi d_{0}/k_{B}T}}{2I_{0}\left(d_{0}/k_{B}T\right)}\int_{1}^{\infty }\frac{e^{-d_{0}E/k_{B}T}}{\sqrt{E+1}\;E\left(\sqrt{2/(E+1)}\right)}dE,$$
where $D_{0}=k_{B}T/\gamma$ is the Einstein diffusion coefficient. Let us note that also this diffusion coefficient does not depend on the Brownian particle mass *m*.

The integral cannot be calculated analytically in a closed form. However, two regimes of low and high temperatures can be evaluated; see Appendix B. In the low temperature limit, $k_{B}T\ll d_{0}$, the asymptotic expression for *D* reads [44,45,46]
(48)$$D∼\frac{k_{B}T}{\gamma }\;e^{-2d_{0}/k_{B}T}\to 0\;for\;T\to 0.$$ In the high-temperature limit, $k_{B}T\gg d_{0}$, the diffusion coefficient tends toward the Einstein’ one,
(49)$$D\to D_{0}=\frac{k_{B}T}{\gamma }.$$ This result appears to be obvious because at very high temperatures, thermal noise surpasses the conservative force stemming from the periodic potential, and the latter can in principle be neglected. Consequently, the particle moves essentially freely.

A comparison of $D=D_{2}$ in Equation (Equation 47) with the Einstein $D_{0}$ and the overdamped diffusion coefficient $D=D_{1}$ in Equation (Equation 34) is presented in Figure 1. We observe that diffusion in the periodic potential is always slower compared with the free particle dynamics. In addition, in the underdamped regime, it proceeds slower than for the overdamped situation, i.e., $D_{2}<D_{1}<D_{0}$. In the limit of low temperature, $D_{2}$ in (Equation 47) approaches zero much faster than in the overdamped regime. Both $D_{1}$ and $D_{2}$ are non-linear but are monotonically increasing functions of the temperature *T*. It should be contrasted with the Einstein model for which *D* is a linear function of *T*.

## 7. Diffusion in Tilted Spatially Periodic Potentials

The next generalization of the model is represented by the Langevin equation of the form
(50)$$m\ddot{x}+\gamma \dot{x}=-U^{\prime }\left(x\right)+F+\sqrt{2\gamma k_{B}T}\xi \left(t\right),$$
where now the constant force *F* acts additionally to the periodic force $F\left(x\right)=-U^{\prime }\left(x\right)$. The effective potential
(51)U(x)=U(x)−Fx
is known as the *washboard potential* or the tilted periodic potential. It is an example of a *nonequilibrium system* possessing a stationary state in which transport is generated by thermal fluctuations. Similarly to the previous section, there are two limiting regimes, overdamped and full inertial dynamics, that need to be treated separately.

### 7.1. Overdamped Dynamics

The mass *m* of the Brownian particle formally set to the vanishing value m=0 yields, for the dynamics the following overdamped Langevin equation,
(52)$$\gamma \dot{x}=-U^{\prime }\left(x\right)+F+\sqrt{2\gamma k_{B}T}\xi \left(t\right).$$ This is one of the simplest systems that can render a non-monotonic temperature dependence for the diffusion coefficient *D*; i.e., there are parameter regimes in which *D* decreases when temperature is increased. Such behavior is counter-intuitive, and it clearly also contradicts the Einstein relation for the diffusion coefficient $D_{0}=k_{B}T/\gamma$, which displays a strictly monotonically (linearly) increasing function of the environment temperature *T*.

In 2001, two groups presented equivalent formulas for the diffusion coefficient [47,48,49]. In Reference [47], the diffusive motion of an overdamped particle in a stylized biased periodic potential was analyzed. At a sufficiently strong but subcritical static force, an optimized diffusion with respect to temperature was observed. In References [48,49], a more compact formula for the diffusion coefficient was derived. Their pertinent result reads
(53)$$D=D_{0}\;\frac{E[I_{+}^{2}I_{-}]}{{\left(E[I_{+}]\right)}^{3}},$$
where $D_{0}$ is the Einstein free-diffusion coefficient, E[·] indicates the average over the spatial period *L* of the potential as defined in Equation (Equation 32), and the functions $I_{\pm }\left(x\right)$ are given by the relation
(54)$$I_{\pm }\left(x\right):=\int_{0}^{L}e^{\left\{\pm U\left(x\right)\mp U\left(x\mp y\right)-yF\right\}/k_{B}T}\;dy.$$ In Reference [48], the authors reported that for weak thermal noise and near the critical tilt $F=F_{c}$ (where the deterministic running solutions set in), the diffusion coefficient becomes gigantically enhanced versus the free diffusion $D\gg D_{0}$. This phenomenon was coined as giant diffusion. In such a case the dynamics given by Equation (Equation 52) can be divided into two processes, (i) the particle relaxation towards the minimum of the potential U(x), as well as (ii) thermal noise driven escape from the latter position. The first process is robust with respect to temperature variation, but the escape time is very sensitive to changes of this parameter. This dichotomy lays at the root of the giant diffusion phenomenon. Moreover, in this regime the diffusion coefficient displays a non-monotonic temperature dependence.

Later, with Reference [50] the authors studied the same system but with a spatially dependent friction and reported that, in some parameter regimes, an increase in temperature is accompanied by a decrease in the diffusion coefficient, a fact which physically appears rather counter-intuitive. A similar effect was predicted analytically for a tilted periodic piecewise potential with one and two maxima per period [51,52]. The giant diffusion phenomenon was experimentally verified in 2006 in a setup consisting of a single colloidal sphere circulating around a periodically modulated optical vortex trap [53]. A tilted periodic potential was generated also by means of rotating optical tweezers arranged on a circle [54]. In [55], the authors showed that the presence of weak disorder may further boost a pronounced enhancement over the free thermal diffusion within a small interval of tilt values by orders of magnitude. Further experimental manifestation for the giant diffusion effect includes transport in a tilted two-layer colloidal system [56] and single-molecule study on F-1-ATPase [57]. By using the general Kubo formalism the authors of Reference [58] analytically calculated the full temporal behavior of dispersion of particles diffusing in a tilted periodic potential. A tight-binding approach to overdamped Brownian motion in a biased periodic potential was developed in [59] as well. Recently, a closely related phenomenon of colossal diffusion, drastically surpassing the previously researched situation known as giant diffusion, has been predicted for the overdamped Brownian particle dwelling in the periodic potential and exposed to active nonequilibrium noise [60,61].

### 7.2. Full Inertial Dynamics

In the overdamped regime, the deterministic dynamics of the system displays only a creeping motion; i.e., if the tilted potential exhibits minima, the particle is pinned in one of the potential wells (locked solution), or when the static bias is large enough and the minima cease to exist, the particle slides down the potential (running solution). This picture is significantly changed when the full inertial dynamics governed by Equation (Equation 50) is considered. If the tilted potential exhibits minima, then, due to the finite momentum of the particle, it is possible that the particles can overcome the potential barrier if the damping is sufficiently small [45,62]. Such coexistence of the locked and running solutions for the deterministic dynamics is termed bistability [45,63].

The dependence of diffusion in a tilted periodic potential on temperature in the underdamped regime was considered in Reference [64] for crystalline surfaces under static external forcing. These authors found that the maximal diffusion coefficient $D_{max}$ for a biased system grows when temperature is decreased in a power-law manner $D_{max}\propto T^{-3.5}$, and the force range of diffusion enhancement shrinks to zero when temperature decreases to zero. In Reference [65] it has been shown that $D_{max}\propto T^{2/3}\exp\left(\epsilon /k_{B}T\right)$ with a drop in temperature in a certain interval of the static bias. Depending on the damping strength, diffusion either tends to zero or increases. Next, in Reference [66], the authors considered the problem by converting dynamics to the velocity space with an effective double-well potential. An approximate expression for the diffusion coefficient was derived for this simplified model. Later, in Reference [67], a counterpart of the giant diffusion effect was predicted in the underdamped regime. The authors used a two-state theory to determine for all values of the friction coefficient γ the range of the force $f\in [f_{gd,-},f_{gd,+}]$, in which the diffusion coefficient increases exponentially to infinity as the temperature decreases towards zero. They indicated that outside of this interval, the diffusion *D* possesses a pronounced maximum as a function of temperature, and it diminishes exponentially to zero for T→0. The width of the region of giant enhancement of diffusion was found to be a non-monotonic function of the friction coefficient γ, exhibiting a distinct maximum. In Reference [68], it has been found that the force interval of the temperature abnormal behavior decreases linearly with an increase of γ, whereas the diffusion coefficient in this range increases linearly with γ. An analytical expression have been obtained for the diffusion coefficient in the low-temperature limit.

The non-monotonic behavior of the diffusion coefficient was also detected under the additional presence of nonequilibrium Ornstein–Uhlenbeck noise [69]. In References [70,71], the authors revisited the problem and constructed a phase diagram for the occurrence of a non-monotonic dependence of the diffusion coefficient on temperature, which extends the original parameter region related to the giant diffusion phenomenon. The weak noise limit of diffusion in this system was re-investigated in Reference [72], where, in contrast to previous results on this topic, it was shown that in the parameter regime where the bistability of solutions is observed, the lifetime of ballistic diffusion diverges to infinity when the temperature approaches zero; i.e., an everlasting ballistic diffusion emerges. Consequently, the diffusion coefficient does not reach its stationary constant value. Recently, a new platform in the form of the rotational motion of a nano-dumbbell driven by an elliptically polarized light beam [73] was proposed to study this limit experimentally.

From the above discussion, it follows that diffusive motion in a tilted periodic potential is much more complex than it might seem at first glance. In some regimes, diffusion is normal, and the diffusion coefficient assumes finite values. In some parameter regions, the time of transient anomalous diffusion approaches infinity, and consequently the diffusion coefficient cannot be defined. In Figure 2, we present an exemplary temperature dependence of the diffusion coefficient in the regime in which diffusion is normal. In the overdamped case, we use the dimensionless Langevin Equation (Equation 69), while for the full inertial dynamics Equation (Equation 62) is applied. In the overdamped regime there is a temperature interval where the diffusion coefficient is greater than the Einstein value $D>D_{0}$. For higher temperature, $D<D_{0}$ and it approaches $D_{0}$ from below as temperature increases. For the full inertial dynamics, the diffusion coefficient can assume values much greater than in the overdamped limit. In the regime of velocity, the bistability *D* first decreases as temperature grows, reaches a minimum, and then tends to the Einstein’s value $D_{0}$. The simplified explanation of this behavior is the following [71,72]. There are two contributions to *D*: (i) the spread of trajectories between the running and locked solutions and (ii) the spread of trajectories inside both states. At very low temperatures, the first contribution is dominant, and therefore *D* is large. If the temperature increases, the lifetimes in both states becomes shorter and shorter, and the particle jumps more frequently between both states. Consequently, the contribution (i) is smaller and *D* decreases. The minimum of *D* occurs when the stationary probability for the particle to reside in the locked state is minimal. The impact of further increases in temperature is similar to that of the Einstein mechanism, i.e., growing thermal fluctuations cause an increase in *D*.

The clear distinction between overdamped and inertial regimes is observed when temperature drops to zero. In such a case, the diffusion coefficient in the overdamped limit always tends to zero, whereas for the inertial dynamics, the diffusion is ballistic; i.e., its coefficient is not defined, provided that the given parameter regime exhibits the velocity bistability—for details, see Reference [72].

## 8. Diffusion in Time-Periodic-Driven Spatially Periodic Systems

As the last generic setup, we consider a Brownian particle moving in a spatially periodic potential U(x) and simultaneously being subjected to an external, unbiased, time-periodic forcing acos(ωt) of angular frequency ω and amplitude strength *a*. The respective inertial Langevin dynamics assume the form
(55)$$m\ddot{x}+\gamma \dot{x}=-U^{\prime }\left(x\right)+a\cos\left(\omega t\right)+\sqrt{2\gamma k_{B}T}\;\xi \left(t\right).$$ The rich deterministic physics contained in this model has become evident in recent decades with numerous studies. In particular, with time-periodic driving, this class of systems comprises operational regimes that are deterministically chaotic. The sensitive dependence on initial conditions and the abundance of unstable periodic attractors are the most salient characteristics of chaotic behavior [74]. The combination of these features, possibly assisted with additional noise agitation, makes this system one of the most flexible setups enabling the emergence of different peculiar behavior. Depending on the parameter values, its deterministic counterpart (i.e., when thermal noise term is set to zero) displays periodic, quasi-periodic, and chaotic motion. However, the price for this variety is the absence of any analytical solutions for this class of models. Moreover, in contrast to previous examples, in this case, the long-time state of the system is (i) in nonequilibrium and also (ii) non-stationary.

### 8.1. Symmetric Systems

When the potential is reflection-symmetric, i.e., there exists a shift $x_{0}$ such that $U\left(x_{0}+x\right)=U\left(x_{0}-x\right)$, the mean long-time particle velocity vanishes identically, since all terms in the right hand side of Equation (Equation 55) are symmetric and of zero-mean.

The first study of *inertial* Brownian motion of a time-periodically driven Brownian particle in a periodic potential is that of Jung and Hänggi in 1991 [75]. Therein, upon using full Floquet theory of the underlying Fokker–Planck operator, the authors addressed the topic of time-dependent driven-escape rates. These latter knowingly also determine the diffusion coefficient [75,76]. The diffusion coefficient of overdamped Brownian particle driven by an adiabatic slow time-periodic force and moving in a sinusoidal periodic potential has been studied by Gang et al. in Reference [77]. The authors found that the diffusion in the overdamped regime may exceed the free thermal diffusion for optimal parameter matching. Moreover, it can display a non-monotonic temperature dependence. Using a stylized piecewise linear periodic potential, the feature of *oscillations* of the diffusion coefficient as a function of (stepwise) time-periodic driving was reported in Reference [78].

With the work [79], the authors studied the coherence of the transport of an overdamped Brownian dwelling in a sinusoidal potential and driven by an unbiased temporally asymmetric time-periodic force. This system exhibits giant coherence of transport measured by the Peclet number in the regime of a parameter space where unidirectional currents in the deterministic case are observed. The transport coherence, as well as the diffusion coefficient, can render the non-monotonic temperature dependence. Oscillations of the diffusion strength for a periodically driven particle suggest the potential for an efficient scheme in separating matter at the submicron scale. This has been investigated in the literature in Reference [80]. The non-monotonic diffusion as a function of temperature has similarly been found also for a two-dimensional system consisting of an overdamped particle moving on a square lattice potential in the presence of externally applied AC driving [81]. For sufficiently small temperatures, the diffusion along a given axis can become arbitrarily large, whereas for a suitably chosen second axis, it tends to vanish, thus providing a tool for a enormous enhancement and control.

Numerical studies of the diffusion coefficient in the full inertial regime has been considered in 2012 with Reference [82]. The authors detected an increase in diffusion when the temperature of the system is lowered and reported that for appropriate parameter regimes, it can be orders of magnitude greater than the free thermal diffusion. Likewise, with Reference [83], the authors reported a situation in which the diffusion coefficient decreases with increasing temperature within a finite temperature window as well. A simplified stochastic model was there formulated in terms of a three-state Markovian process, which allowed us to explain the mechanism ruling out this counterintuitive effect. It is rooted in the deterministic dynamics consisting of a few unstable periodic orbits embedded into a chaotic attractor together with thermal noise-induced transitions upon varying temperature. The impact of an external time-periodic driving on the emergence of non-monotonic temperature dependence of diffusion has been addressed in [84]. The authors found that at any fixed driving frequency, diffusion is an increasing function of temperature, provided that the temperature is sufficiently low. Recently, the problem of diffusion has been revisited again [85]. The authors revealed further parameter domains in which diffusion is normal in the long time limit and exhibits intriguing giant damped quasiperiodic oscillations as a function of the external driving amplitude. As the mechanism behind this effect, they identified the corresponding oscillations of difference in the number of locked and running trajectories, which carries the leading contribution to the diffusion coefficient. Experimental realizations of an driven inertial Brownian particle in a periodic potential involve cold atoms in optical lattices [86], or also colloidal particles placed in light fields [87], to mention a few.

In Figure 3, we display a distinctive non-monotonic dependence of the diffusion coefficient on temperature. The rough explanation of this behavior is the following [83]. In the presented parameter region for a deterministic system, there is a locked solution and many unstable periodic orbits (running states), which move in pairs in the opposite direction. Thermally induced transitions between various states change the populations of certain regions in phase space. When the probability $p_{r}$ of staying in the running states decreases with temperature and at the same time the probability $p_{l}$ being in the locked state increases, then the diffusion coefficient *D* decreases. If the difference $p_{r}-p_{l}$ is maximal, then *D* attains its maximum; when it decreases, *D* diminishes as well and approaches a minimum when $p_{r}=p_{l}$. At sufficiently high temperature, the population of various orbits is almost homogeneous, and as temperature continues to rise, this causes a monotonic increase in the diffusion coefficient *D*.

### 8.2. Ratchet Systems

For ratchet systems, the reflection symmetry of the spatially periodic potential is broken, implying that $U\left(x_{0}+x\right)\neq U\left(x_{0}-x\right)$ for all shifts $x_{0}$. In such a case, the breaking of the detailed balance symmetry induced by the driving acos(ωt), takes the system out of equilibrium, is sufficient to generate a directed transport even in the absence of any biased external forces of deterministic or stochastic nature [88,89].

Despite many years of intense and beneficial research in ratchet physics, an analysis of thr diffusion anomalies in such systems has been addressed only very recently [90]; those authors studied the dynamics of an inertial Brownian particle in an asymmetric periodic potential while driven by external harmonic driving. They discovered a diversity of unusual diffusive effects, including various regimes of transient anomalous diffusion and also diffusion suppressed by thermal noise in which a normal diffusion coefficient exhibits non-monotonic dependence on temperature. The former has been analyzed in detail in a series of subsequent papers [91,92,93], and the mechanism of the latter phenomenon has been explained in Reference [94]. The latter effect originates from the temperature dependence of transitions between regions in the phase space dynamics of the particle. Several examples of experimental realizations of a driven ratchet setup are cold atoms in optical lattices [95], the Josephson phase difference in SQUIDs [96], and nanofluidic rocking Brownian motors [97].

As an example, let us consider the stochastic dynamics of an inertial Brownian particle in a ratchet potential of the form [94]
(56)$$V\left(x\right)=-\sin x-\frac{1}{4}\sin2x.$$ The system dynamics is modeled by Equation (Equation 67) from Appendix A. In the parameter regime M=6,A=1.899,Ω=0.403 and f=0, the deterministic counterpart of the setup, i.e., for θ=0, is non-chaotic and possesses three attractors with velocities $v_{+}\approx 0.4,v_{0}\approx 0,v_{-}\approx -0.4$. There are three corresponding classes of trajectories: x(t)∼0.4t, x(t)∼0, and x(t)∼−0.4t. When thermal fluctuations agitate the stochastic system, the dynamics destabilizes the attractors and generates random transitions among them. Since the symmetry of the potential is broken, the directed transport emerges in the long time limit with the positive averaged velocity 〈v〉≈0.4. If temperature grows, the jumps between different types of solutions occur more frequently; then, the mean time to destroy the deterministic structure of attractors becomes shorter and 〈v〉 starts to decrease.

In Figure 4, we display the dependence of the diffusion coefficient *D* on temperature θ∝T. *D* behaves there in a non-monotonic manner, similarly as in Figure 3. For low temperatures, *D* initially increases, passes through its local maximum for θ≈0.0045, and next starts to decrease, reaching its minimum for θ≈0.76. At larger temperatures, *D* monotonically increases and becomes strictly proportional to θ. The analysis presented in Reference [94] evidences that for such multistable velocity dynamics, there are three contributions to the spread of trajectories of the particle and consequently to the diffusion coefficient *D*. A first one, which is the leading one, stems from the spread associated with the relative distance between the locked $V_{0}$ and the running $\left\{V_{-},V_{+}\right\}$ trajectories. The second and third parts correspond to a thermally driven spread of trajectories, following the locked and running solutions, respectively. At the maximum of *D*, the averaged velocity 〈v〉 is also large, and the first contribution dominates. If the temperature increases, this contribution becomes reduced because 〈v〉 also decreases, and at the minimum of *D*, the particle frequently jumps between three states with 〈v〉≈0. At a high temperature limit, the population of these three classes of trajectories is homogeneous, thus leading to the Einstein-type behavior of the corresponding diffusion coefficient.

## 9. Discussion and Sundry Topics

In this review, we have discussed the rich behavior of the diffusion coefficient of a Brownian particle dynamics in equilibrium and nonequilibrium states. We started with the spreading of a free Brownian particle for which we presented in detail the approaches by Albert Einstein as well as by his contemporaries, i.e., William Sutherland, Marian Smoluchowski, and Paul Langevin, who all performed pioneering work for this most central and salient physical phenomenon. We then demonstrated in a step-by-step manner how the increase in model complexity affects these original theories of normal diffusion. In doing so, we first considered the impact of a nonlinear spatially periodic potential for the diffusion of Brownian particles. It turned out that in equilibrium, the diffusion coefficient is always reduced compared to the free particle; however, it is still an increasing function of temperature for the system. This behavior is changed significantly when the particle is additionally subjected to a constant bias, which in turn drives the stochastic dynamics out of equilibrium towards a nonequilibrium stationary state. Such a setup can exhibit a nonmonotonic temperature dependence of the diffusion coefficient, which clearly contradicts the conventional theory. This effect also emerges when the Brownian particle moves in a periodic potential and is driven by an unbiased time-periodic force, which takes it away from thermal equilibrium into a nonequilibrium and time-dependent state. Given the apparent simplicity of underlying dynamics formulated in terms of an overdamped or also fully inertial Langevin equation for a single Brownian particle, we find that these models are minimal for the emergence of an intriguing non-monotonic temperature dependence in various regimes of diffusion. This non-monotonicity of *D* vs. *T* not only is a theoretical prediction but has been detected in many systems. Probably the first experimental manifestation was the diffusion of nickel atoms in austenitic chromium–nickel steels at high-speed deformation [98]. Ten years later, others observed this non-monotonic temperature dependence of the diffusion of helium in a 3He−4He solid mixture [99]. It was also shown that grains of titanium can grow more intensively when temperature decreases [100]. Diffusion along grain boundaries in Al-based alloys can exhibit a remarkable nonmonotonic dependence on the annealing temperature [101]. Notably, such peculiar diffusive behavior has been detected not only in solid-state experiments but also for soft-matter setups such as liquid crystals [102] and coupled protein diffusion in the cell [103], and also for hydroxide ion diffusion in anion exchange membranes [104]. There exist other examples in which this counter-intuitive feature has been identified, such as diffusion in quantum disordered systems [105] and spreading of quantum excitations [106] or spins [107].

In writing this review, we focused on the diffusion of Brownian particles in equilibrium and nonequilibrium states. To keep the article as simple as possible and accessible to a broad range of readers, we limited ourselves to a one-dimensional dynamics described by the Langevin stochastic differential equation. We considered the temperature dependence of the diffusion coefficient for a Brownian particle moving freely or dwelling in a periodic potential. Closely related topics of ongoing research are not reviewed here. We did not cover diffusion on surfaces [108,109], confined geometries [110,111], or disordered media [112,113,114]. Moreover, we restricted ourselves to normal diffusive behavior, meaning that the mean-square displacement of the particle is a linearly increasing function of the elapsed time. The deviation from this rule, known as anomalous diffusion [115], is another field of ever active research. Subdiffusion [116,117,118] emerges when the mean square displacement of the particle scales with time slower than for normal diffusion, whereas superdiffusion [119] occurs if the particle spreading is faster than in the linear case. Anomalous diffusion is routinely detected in the crowded world of biological cells, where stochastic models are applied to model intracellular transport far from thermal equilibrium [120,121,122]. Experimental progress in high-energy physics and astrophysics has stimulated the unification of relativistic and stochastic concepts and has led to the development of the theory of relativistic Brownian motion [123]. We did not cover the diffusion of active particles [124,125,126], also known as self-propelled Brownian particles, which are capable of taking energy from their environment and converting it into directed motion. Diffusion in more complex systems such as hot hadronic matter [127], quark–gluon plasma [128], solids, polymers, gels, glasses, and supercooled melts [129,130] lies beyond the limited scope of the current work. It is no different for diffusion on complex networks [131,132] or in brain tissues [133], nor diffusion of innovations in service organizations [134].

As the last comment, we want to note that the Einstein free-diffusion coefficient given in Equation (Equation 6) constitutes a particular manifestation of the so-called Einstein relation
(57)$$D=\tilde{\mu }k_{B}T,$$
where $\tilde{\mu }=\lim_{F\to 0}\;\langle v\rangle /F$ is the mobility of the particle in the linear response regime, i.e., the ratio of its average velocity 〈v〉 to the applied constant force *F*. For a free Brownian particle, one can easily find that $\tilde{\mu }=1/\gamma$ and Equation (Equation 57) reduces to the expression (Equation 6). The Einstein relation links the diffusion coefficient *D*, a property of the unperturbed system, and the mobility $\tilde{\mu }$, which measures the system response to a small perturbation. It is a consequence and one of the examples of the fluctuation-dissipation theorem [22,26,27,28], which is valid for systems in thermal equilibrium with a perturbation applied in the linear response regime. On one hand, the calculation of the mobility $\tilde{\mu }$ as a function of system parameters is not an easy task in general. As an example, we mention the many-aspect issue of diffusion of non-spherical molecules near confining surfaces in which hydrodynamic interactions with boundaries introduce an additional, anisotropic drag acting on molecules, and the diffusion coefficients of arbitrarily shaped bodies become complicated functions of their position and orientation relative to surfaces [135]. Another example concerns inertial effects with strong implications for biophysics and molecular biology [136,137]. On the other hand, there are attempts to generalize the Einstein relation beyond equilibrium [138,139,140,141,142,143] and for complex setups such as disordered systems [144], aging colloidal glasses [145], supercooled liquids [146], and nanoparticle diffusion in polymers [147], to mention only a few.

To conclude, this review serves as a bridge between history and the state of the art of the diffusion coefficient of a Brownian particles both in and out of equilibrium. We demonstrated that more than 100 years after pioneering works by Sutherland, Einstein, Smoluchowski, and Langevin, the founding fathers of modern statistical physics, it is still in the spotlight of physics attracting even researchers coming from other sciences. As our exploration of the microworld is not completed yet, and with nanotechnology nowadays it paradoxically accelerates, we are sure that diffusion will continue to play a leading role in our understanding of microscopic reality and *“there is a plenty of room”* to be discovered in this field of research.

## Figures and Tables

**Figure 1 entropy-25-00042-f001:**
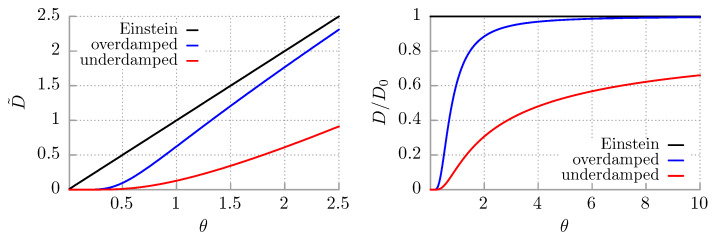
Temperature dependence of the diffusion coefficient is depicted for three cases: the Einstein form (Equation 6) for a free Brownian particle, in the overdamped (Equation 34) and underdamped (Equation 47) regimes for the Brownian particle moving in the periodic potential $U\left(x\right)=-d_{0}\cos\left(x\right)$. In the left panel, the approach to zero temperature is displayed, while in the right one, the high temperature region is visualized. The dimensionless diffusion coefficient is $\tilde{D}=\gamma D/d_{0}$, and the dimensionless temperature reads $\theta =k_{B}T/d_{0}$.

**Figure 2 entropy-25-00042-f002:**
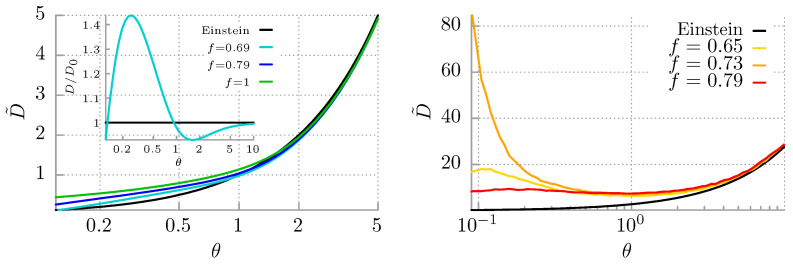
Diffusion in the tilted periodic potential U(x)=−sin(x)−Fx. Left panel: The diffusion coefficient $\tilde{D}$ versus temperature θ is shown for different bias values in the overdamped regime; c.f. Equation (Equation 69). Right panel: the same characteristics depicted for the full inertial dynamics with Γ=0.4; c.f. Equation (Equation 62).

**Figure 3 entropy-25-00042-f003:**
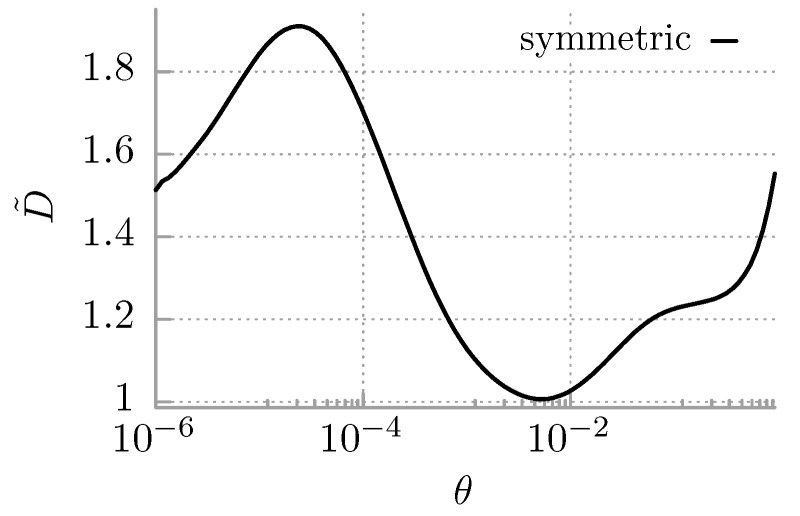
Brownian motion in the symmetric potential V(x)=sin(2πx) and driven by the time-periodic force Acos(Ωt). The dimensionless diffusion coefficient $\tilde{D}$ versus temperature θ is visualized. The parameters in Equation (Equation 67) are M=0.9, A=8.699997, Ω=0.2754226,f=0.

**Figure 4 entropy-25-00042-f004:**
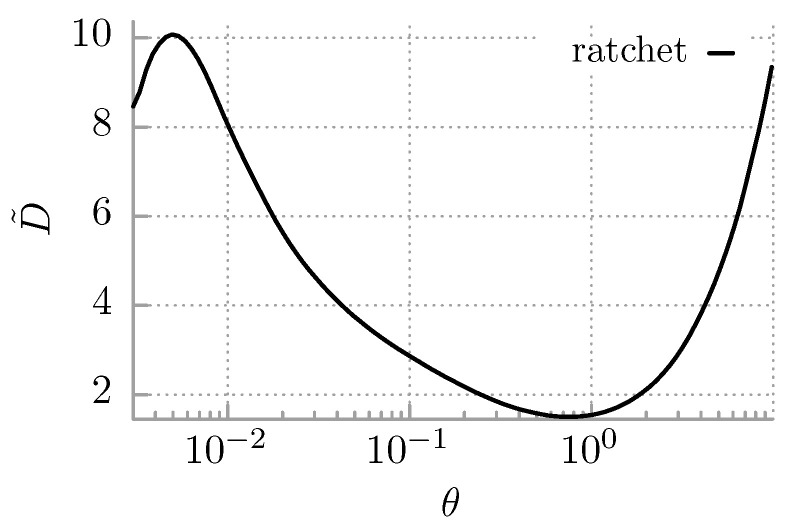
Diffusion in the ratchet potential (Equation 56) and driven by the time-periodic force Acos(Ωt). The dimensionless diffusion coefficient $\tilde{D}$ versus thermal noise intensity θ is depicted. Parameters in Equation (Equation 67) are M=6,A=1.899,Ω=0.403,f=0.

## Data Availability

The data that support the findings of this study are available from the corresponding author upon reasonable request.

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
