# Peer review of "Diffusion Coefficient of a Brownian Particle in Equilibrium and Nonequilibrium: Einstein Model and Beyond"

_entropy, 2022, doi:10.3390/e25010042_

Round 1
Reviewer 1 Report
This is a review of the manuscript by Jakub Spiechowicz, Ivan G. Marchenko, Peter H\"anggi and Jerzy Luczka titled ``Diffusion coefficient of a Brownian particle in equilibrium and nonequilibrium: Einstein model and beyond.'' The authors review diffusion processes from the Einstein model to diffusion in tilted potentials. Diffusion processes are important issues in statistical physics. Overall my impression, the manuscript is written very well and the recent results are also mentioned. Therefore, I recommend its publication.
Reviewer 2 Report
The manuscript reviews results on the diffusion of a Brownian particle modeled with Langevin dynamics, with particular attention to the temperature dependence of the diffusion coefficient.
It starts with a fascinating historical introduction. Many interesting results are clearly reported in the review from the beginning of the manuscript to Subsection 7.1. However, results reported on full inertial dynamics in Subsections 7.2, 8.1, and 8.2 are difficult to understand with the level of description provided. Further details on the underlying mechanisms are needed for these results.
Comments on mobility in the discussion are interesting. Perhaps it would be interesting to present, describe or explain the results reviewed in the manuscript in terms of mobility. Explaining the large diffusion coefficients as an effect of high mobility regimes, provided the authors consider this approach to be rigorous.
The discussion includes an interesting enumeration of related phenomena and studies, which expand the contents of the present review. Other potentially interesting subjects include flashing ratchets, feedback ratchets, and interacting particle ratchets.
Suggestions for improvements
Section 2 will benefit from a change of notation to a more usual choice of variables. For example, rho for the density of solute molecules (instead of v), F for the Stokes force (instead of K), a for the particle radius (instead of P), eta for the viscosity (instead of k). This notation, closer to the Sutherland notation, is more intuitive for a present reader. (Historical notation can be reported in parenthesis as done in the current version with the Sutherland notation.) (Note that in the submitted version, there are sets of variables with similar letters and quite different meanings: V, v; K, k, k_B; P, p.)
Further discussion of why Lambda is in Eq. (19) could make this part more informative (or at least cite the more appropriate references on the fluctuation-dissipation theorem on this point).
Subsection 7.2: Further details and/or graphical representations on locked and running solutions might help the reader to get a deeper understanding of the full inertial dynamics results. The second paragraph mentions a decrease of the maximum diffusion with the temperature. This counterintuitive result needs an explanation that helps the reader understand the underlying mechanism. The end of this second paragraph is also hard to read. The presence of a maximum and the behavior in the limit of zero temperature are described in the same sentence; split this sentence in two. Try to give more flow to the description and context to the behaviors. These comments also apply to the last paragraph on page 14 and the last paragraph of Section 8.
Typo in page 13: “time-periodic periodic driving”
Please note that there is a typo in Ref. 90. The title is repeated.
